# Indocyanine Green-Assisted and LED-Light-Activated Antibacterial Photodynamic Therapy Reduces Dental Plaque

**DOI:** 10.3390/dj9050052

**Published:** 2021-05-03

**Authors:** Sakari Nikinmaa, Niina Moilanen, Timo Sorsa, Juha Rantala, Heikki Alapulli, Anja Kotiranta, Petri Auvinen, Esko Kankuri, Jukka H. Meurman, Tommi Pätilä

**Affiliations:** 1Department of Neuroscience and Biomedical Engineering, Aalto University, 12200 Espoo, Finland; sakari@koitehealth.com (S.N.); narantala@gmail.com (J.R.); 2Department of Oral and Maxillofacial Diseases, Faculty of Medicine, University of Helsinki and Helsinki University Hospital, 00290 Helsinki, Finland; niina.moilanen@helsinki.fi (N.M.); timo.sorsa@helsinki.fi (T.S.); heikki.alapulli@helsinki.fi (H.A.); anja.kotiranta@helsinki.fi (A.K.); jukka.meurman@helsinki.fi (J.H.M.); 3Department of Oral Diseases, Karolinska Institutet, 14152 Huddinge, Sweden; 4Institute of Biotechnology, University of Helsinki, 00790 Helsinki, Finland; petri.auvinen@helsinki.fi; 5Faculty of Medicine, Department of Pharmacology, University of Helsinki, 00290 Helsinki, Finland; 6Department of Congenital Heart Surgery and Organ Transplantation, New Children’s Hospital, University of Helsinki, 00290 Helsinki, Finland; tommi.patila@hus.fi

**Keywords:** antibacterial photodynamic therapy, dental plaque, gingivitis

## Abstract

Aim: This study aimed to determine the feasibility and first efficacy of indocyanine green (ICG)-assisted antimicrobial photodynamictherapy (aPDT) as activated using LED light to the dental plaque. Methods: Fifteen healthy adults were assigned to this four-day randomized study. After rinsing with ICG, 100 J/cm^2^ of 810 nm LED light was applied to the aPDT-treatment area. Plaque area and gingival crevicular fluid (GCF) matrix metalloproteinase-8 (MMP-8) were measured, and plaque bacteriomes before and after the study were analyzed using 16S rRNA sequencing. Results: aPDT administration was preformed successfully and plaque-specifically with the combination of ICG and the applicator. Total plaque area and endpoint MMP-8 levels were reduced on the aPDT-treatment side. aPDT reduced *Streptococcus*, *Acinetobacteria*, *Capnocytophaga*, and *Rothia* bacteria species in plaques. Conclusion: ICG-assisted aPDT reduces plaque forming bacteria and exerts anti-inflammatory and anti-proteolytic effects.

## 1. Introduction

Dental diseases are among the most common chronic diseases in the world [1,2]. According to the World Health Organization, 60–90% of schoolchildren and most adults worldwide have suffered from caries lesions. Thirty percent of the world population aged 65–74 have no natural teeth. Severe periodontitis, which may result in tooth loss, is found in 5–15% of most populations, while gingivitis affects 50–90% of adults worldwide [1,2]. Importantly, negative impacts resulting from poor oral health are not limited to teeth and the oral cavity but are also associated with the development or worsening of the course of several diseases, affecting the whole body [3].

Dental diseases are mainly a result of pathogenic bacterial colonization on the surfaces of teeth and gingival pockets [4,5]. Mechanical cleaning is the cornerstone of preventive dental care, but antibacterial treatments are commonly used in conjunction to enhance oral hygiene. Optimal preventive antibacterial solution for dentistry should maintain the plaque at levels compatible with oral health, but, in addition, it should have a targeted effect on the biofilm or otherwise preserve the normal oral microbiome, antibacterial resistance formation should be non-existent, and side effects should be absent.

Antimicrobial photodynamic therapy (aPDT), the combination of light and a light-absorbing photoactive substance, is a very effective antibacterial method with no known resistance formation. In the presence of oxygen, it releases energy in the form of singlet oxygen and heat that destroy bacteria, fungi, and viruses [6]. Research concerning aPDT have demonstrated its potent efficacy against dental pathogens, and the relative effectiveness against streptococci makes the approach very appealing [7]. However, aPDT use in clinical dentistry has been limited to in-office treatment [8], and home-use appliances are not readily available. In-office treatment must be applied by a professional, it is pricey and cumbersome, and it eventually restricts the repeated use of the method. Eventually, the most salient elements of aPDT, locally applied effective antibacterial action with the lack of bacterial resistance formation, are missed [9]. Fortunately, recent developments in LED technology have opened the possibility to administer aPDT at home.

Indocyanine green (ICG) is a US Food and Drug Administration-approved photosensitizer. Previously, ICG has been primarily employed in diagnostic approaches, such as assessment of hepatic function and perfusion-related analysis of tissues, due to its non-toxic nature and fast metabolism [10]. Recently, owing to its absorption peak at close to 810 nm diode lasers, ICG has arisen as a promising tool in dentistry. It has been suggested that ICG-mediated aPDT can provide a potential therapeutic benefit in the treatment of chronic periodontitis patients; multiple studies evaluating the effect of ICG as a photosensitizer have found positive results [11,12,13,14,15,16].

The purpose of the present study was to further investigate the early feasibility of repeated aPDT use with 810 nm LED light and ICG photosensitizer in the developing dental plaque. We evaluated the ability of the method to target the plaque by adherence of ICG to plaque, quantified plaque formation as a measure of bacterial load reduction, and used 16S sequencing to measure the repeated aPDT effect on the relative proportion of bacterial taxa genera in the plaque. Finally, we measured aMMP-8 in the gingival crevicular fluid (GCF) as an indicator of early gingivitis.

## 2. Materials and Methods

The study was approved by the ethics committee of the Hospital District of Helsinki and Uusimaa (HUS/827/2018) (on 21 May 2018) and was conducted in accordance with the ethical principles of the Declaration of Helsinki. All participants provided written informed consent before enrolment. The study is registered in the ISRCTN registry (www.isrctn.com, accessed on 29 April 2021) under the trial number ISRCTN36318197. All variables were analyzed blinded to the treatment allocation.

Study design: This randomized, split-mouth study included 15 healthy volunteers. Figure 1A presents the LED applicator and Figure 1B presents the study design. aPDT was administered for a total of four times on four consecutive days using the LED light applicator (Figure 1A) after spurting the ICG photoactive substance. Inclusion criteria included being generally healthy, being between the ages of 18 and 65, and having the ability to refrain from brushing one’s teeth during the treatment period. Exclusion criteria were diabetes, medications that may affect the immune response or saliva secretion, malignancy, pregnancy, dental implants or prosthesis, fixed orthodontic appliances, or active or chronic oral infection including gingivitis or active caries. Medications that could affect the oral microbiome, such as antibiotics, and antimicrobial mouthwashes were not permitted during the study or within the two weeks prior to the study. Decayed, Missing, and Filled Teeth (DMFT) indexes were not collected. Two days before the study, a meticulous professional cleaning was performed on maxillary first premolars on both sides. The treatment side was randomized by coin flip, and the contralateral side served as the control. Study subjects were not allowed to brush or otherwise clean their teeth during the study period.

aPDT application: ICG powder (Verdye, Diagnostic Green GmBH, Aschheim, Germany) was dissolved in water at a weight/volume ratio of 7 mg/25 mL, and each subject spurted the solution for 60 s in their mouth prior to the light application. Near-infrared imaging was performed to ensure the localization of the ICG in the dental plaque (Figure 1C,D). The LED applicator included 16 0.5 W LEDs in a lollipop form, optimally located for producing an even light distribution to the teeth’s surface. Light application time was determined by the target light dose of 100 J/cm^2^, and the treatment time was set accordingly for 8 min per session. Light intensity was decreased if the person under examination felt that the heat of the light applicator was too intense. In such case, the reduction in light intensity was adjusted by an increase in treatment time to regulate the target dose. This adjustment was performed with a light power meter (Thorlabs PM 100D with S121C sensor head, Thorlabs Inc., Newton, NJ, USA).

Plaque analysis: The premolar teeth were photographed daily on both sides of the maxilla using a ProDENT PD740 dental camera (Venoka USA Inc., Windermere, FL, USA) under near infrared and white light lighting conditions to assess ICG adherence to the dental plaque. Daily imaging using a SoproCare camera (Acteon Group Ltd., Norwich, UK) was carried out to assess the development of plaque formation. The final plaque imaging was performed with the ProDENT PD740 camera after plaque staining with an erythrosine tablet according to the manufacturer’s instructions. The amount of dental plaque surface area was measured from the images of both adjacent upper premolars using Photoshop CC software (Adobe Inc., San Jose, CA, USA). On each side, the amount of plaque was determined as the plaque area pixels divided by the total maxillary premolar teeth area pixels.

GCF collection and MMP-8 analysis: GCF samples were collected from the first premolar teeth on each side of the maxilla. Collection of GCF and measurement of clinical parameters were performed prior to any treatment measures, daily before the treatment, and after the last treatment. In total, 72 × 2 GCF samples were collected from the treatment and control sites. GCF sampling was performed by inserting a PerioPaper strip (Oraflow Inc., Hewlett, NY, USA) into the orifice of the gingival sulcus. Samples were collected at the buccal surface, with the insertion point only minorly changing with every sample. GCF sampling was done carefully, and it did not remove any observable amount of supra gingival plaque. Strips contaminated by blood were discarded. The samples were stored in small aliquot containers and kept at −20 °C until analysis. MMP-8 levels were determined by a time-resolved immunofluorescence assay (IFMA) as described previously [17,18,19,20].

16S rRNA bacterial samples: Plaque samples were obtained with Iso Taper Paper Points, size-20 (VDW GmbH, Munich, Germany), by scrubbing the plaque on the tooth enamel. In all the plaque samples, the plaque was intact at the gingival boundary. Each sample was taken from same site by horizontally placing the paper point above gumline of premolar teeth. The paper points were placed into sterile, small-aliquot containers, and were immediately stored at −20 °C until analysis. 16S rRNA sequencing was performed as described earlier [21]. The V3-V4 regions of the 16S rRNA genes were amplified using universal bacterial primers and sequenced with an Illumina MiSeq sequencer (Illumina Inc., San Diego, CA, USA). Bioinformatic analysis, including operational taxonomic unit (OTU) clustering and taxonomy assignment were done using Mothur software [22]. The 16S rRNA sequences have been deposited to the Sequence Read Archive (SRA) of the National Center for Biotechnology Information (NCBI) under BioProject ID PRJNA661546.

Statistical analysis: Sample size calculations were carried out with SAS 9.4 for Windows TS Level 1M4 software (SAS Institute, Cary, NC, USA) power-procedure’s twosamplewilcoxon-statement using an alpha-error level of 0.05 to attain 90% statistical power. The calculated number of participants needed in the split mouth was 10. Eventually, we decided to enroll 15 participants for possible dropouts. Statistical analyses were performed using GraphPad Prism 6.0 (GraphPad Software Inc., San Diego, CA, USA). Unpaired comparisons between the groups were performed with the Mann–Whitney test. The paired rank-sum Wilcoxon test was used for the paired samples. Two-way ANOVA was used to compare the daily values of MMP-8 between groups. Comparisons of categorical variables were performed using Fisher’s exact test. Data are presented as mean ± SEM unless otherwise specified. A *p* value less than 0.05 was considered statistically significant.

## 3. Results

Thirteen subjects completed the entire study protocol. One dropout occurred two days into the study due to a non-related acute infection and another four days into the study, just after the 16S sampling and just before the plaque imaging, due to a work-related emergency. All subjects complied with the inclusion and exclusion criteria. The age of the subjects ranged from 23 to 48, with a mean of 27. Seven were female and eight male. All study subjects were non-smokers. The study subjects were healthy, had no history of deranged saliva secretion, and were periodontitis-free.

### 3.1. Dental Plaque Formation

All subjects were plaque-free at the beginning of the study period. ICG was demonstrated to attach selectively to the dental plaque (Figure 1C). Sopro chromatic mapping did not show any difference in the development speed of the plaque between the aPDT-treatment and control sides. At the end of the study, erythrosine-based plaque-enclosing imaging showed significantly less plaque on the aPDT-treated premolars than controls (35.1 ± 4.8% vs. 42.5 ± 4.0% of the surface area; *p* = 0.016; Figure 2).

### 3.2. Gingival Crevicular Fluid MMP-8 Levels

At the beginning of the study, two days after meticulous professional cleaning, GCF MMP-8 concentrations on both sides were similar. Evaluation of daily pre-treatment MMP-8 concentrations throughout the study using an area under curve analysis demonstrated a trend for lower overall MMP-8 concentrations in the treated group (AUC 256.7 + 99.7, 95% CI 61.2–452.1) as compared to the control side (AUC 344.9 + 113.2, 95% CI 123.0–566.8). However, the GCF MMP-8 levels at the end of the study after aPDT administration demonstrated a significant reduction on the aPDT-treated side (Figure 3A).

### 3.3. Microbiome Analysis and Alpha Diversity

Fourteen paired pre- and posttreatment 16S samples were collected. One sample was disqualified due to low sample quality, because it showed significantly lower bacterial counts together with a significantly reduced amount of OTUs detected. The number of streptococcal species on the aPDT-treated side (median 7460, range 6734–23,987) was significantly reduced as compared to the control side (median 12,776, range 4959–38,669, *p* = 0.0024). Similarly, reduced numbers of *Rothia* species (median 13, range 3–908 vs. median 82, range 3–2883, *p* = 0.0032), *Capnocytophaga* species (median 1050, range 11–4382 vs. median 1220, range 15–8599; *p* = 0.0402) and Actinomyces species (median 141, range 16–1833 vs. median 933, range 12–3608, *p* = 0.0005) were detected on the aPDT-treated vs. control sides. Increases in the OTU-amounts of *Neisseria* (median 7211, range 201–22,892 vs. median 2169, range 122–11,792, *p* = 0.0327), *Haemophilus* species (median 7599, range 713–28,727 vs. median 2935, range 524–27,988, *p* = 0.0479), and *Leptotrichia* (median 307, range 1–5524 vs. median 105, range 4–3431, *p* = 0.0105) were found on the aPDT-treated vs. the control side (Figure 3B).

## 4. Discussion

In this study, we investigated the early feasibility of repeated ICG and 810 nm LED aPDT to improve oral hygiene. We used a randomized split-mouth protocol with restricted oral hygiene to estimate the rough effect of the treatment on a developing dental plaque. During the four days of development, the species in the plaque can be considered to generally consist of cariogenic bacteria, especially in the healthy population studied. ICG, the photosensitizer, attached readily to the plaque. The amount of plaque was reduced on the treated side. In addition, there was a relative reduction in the streptococci in the plaque, with a preserved ecological diversity of the bacterial flora. Moreover, lower amounts of gingivitis and periodontitis biomarker MMP-8 were seen on the treated side. The split-mouth study protocol bolstered the impact of the results, as it minimized inter-individual variability. Particularly in the dental plaque assessment, the diet and the oral microbiome remained the same within the studied mouth, reducing confounding factors.

Dental plaque, a form of bacterial biofilm, is developed by microbes assembled in the extracellular matrix [23]. Antibacterial PDT is especially efficient against Gram-positive bacteria. This is due in part to the structural difference among bacterial species, as well as the mechanism of the treatment [24]. The photosensitizer initiates an electron transfer reaction, activating nearby oxygen molecules to form reactive oxygen species (ROS). The ROS action is eventually bactericidal [7]. Streptococcal species lack catalase enzyme, which is the most important intracellular enzyme in the protection against ROS, rendering the species especially vulnerable [24]. Streptococci are key players in early plaque development [25], and the antibacterial capacity of aPDT provides a means to interfere with early plaque formation.

In addition to its antibacterial effect, aPDT can disturb the structure of the biofilm extracellular matrix, including the enamel pellicle. The main action of the aPDT may be the ability to damage protein structure [24]. The pellicle, being mostly formed by negatively charged glycoproteins from saliva, enables the indocyanine green to bind to the pellicle proteins by electrostatic interaction [10,26]. Although not actually removing the pellicle, it can deform the proteins and derange the ability of bacteria to adhere to the pellicle. Whichever the reason, we found a significant reduction in the early plaque formation on the aPDT treated side. Similar results have been published earlier in a study by Izumi’s group, in which toluidine blue was used as the photosensitizer [27].

We were able to demonstrate plaque-adhering properties of indocyanine green. The adherence of ICG to the dental plaque is partly dependent on the non-covalent bonding of the ICG to the extracellular matrix structure via hydrophilic-hydrophobic forces, hydrogen bonding with polysaccharides within the biofilm matrix; and the electromechanical adherence to the electrically charged bacterial cell membrane. This adherence allows the targeting of most of the treatment effect directly to the dental plaque. Treatment targeting highly reduces possible side effects, in addition to the fact that mammalian cells are much more resistant to the ROS produced by the aPDT than are most procaryotes [24]. Finally, the accumulation of the photosensitizer onto the plaque and focusing of the light onto the same area leaves the other parts of the mouth unaffected. The ability to leave the general bacterial oral flora intact might have a positive impact on promoting a healthy microbiome, bearing in mind that the gastrointestinal tract originates at the mouth. This could be especially important when the aPDT was used in a preventive manner, and the general diversity of mouth bacterial flora is healthy.

The near-infrared 810 nm light offers several advantages. This wavelength has good penetration into tissues [28], which enables the light to penetrate the gingival tissue. The 810 nm wavelength excites human mitochondrial chromophores [29] and increases mitochondrial membrane potential and biogenesis signaling [30,31]. This might be the underlying mechanism in cases in which the 810 nm light has been shown to improve dentin bone formation. The biomodulating effects of the 810 nm infrared light promotes human dental pulp stem cell enhancement, leading to dentinogenesis in vitro [32]. In orthodontic patients with braces, there is also evidence that tooth movement could improve, and pain could be reduced when infrared light is applied [33,34].

Due to the high precision and sensitivity of the 16S rRNA sequencing method, we were able to analyze low abundant genera, and we found additional reductions in capnocytophagal, acinetobacterial, and rothia bacterial groups. These low pathogenicity species have been associated with good health, but, under certain conditions, they participate in the periodontal disease process through several different mechanisms [35,36]. When changes in the species composition in 16S analysis are relative to each other, some species must relatively increase when other species are necessarily reduced. We found the neisseria group to be relatively increasing in the plaque on the treated side. Neisseria has been associated with healthy oral flora, with a correlation to having a younger age, having a lower body mass index, having fewer caries, and being a non-smoker [37]. In addition, the *Haemophilus* species were found to increase in the aPDT treated dental plaque. *Haemophilus* has been shown in several studies to be endowed with healthy periodontal status [38], and larger amounts of *Haemophilus* counts have been found in shallower periodontal pockets [39]. *Leptotrichia* has been identified as part of the normal oral flora.

Dental plaque eventually becomes a structurally and functionally organized community of bacteria. Once formed, aging or temporal maturation of the biofilm changes the bacterial species composition as well as the stability and balance among the species. Importantly, the change in microbial homeostasis is determined by microbial intercommunication and environmental changes. *Capnocytophaga* and *Acinetobacteria* can be associated with periodontitis, and streptococcal species, especially the *S. mutans* strain, are associated with the dental decay process. However, despite having this knowledge of the change in specific bacterial species, it may be difficult for one to estimate the true impact of the change. The total change in the species composition and the reduced ability of plaque formation might have a greater impact. The fact is that the antibacterial impact is located at the supragingival plaque, which is itself undesirable and should be mechanically removed. The question of having plaque disturbance resulting in collateral damage to positively associated bacteria, such as other streptococcal species, has less meaning, especially when the antibacterial treatment is specifically targeted towards teeth. Antibacterial ROS itself is widely in use in nature, including in the bacterial elimination of human white blood cell phagocytosis.

To analyze the effect of the treatment on early gingivitis, we performed an MMP-8 analysis. Matrix metalloproteinase-8 has been well established in the forefront of identification of an active periodontitis process. MMP-8 is especially activated in extensive collagen degradation and osteoclast activation and has been extensively validated [21]. In this study, we found significantly higher concentrations of the MMP-8 enzyme on the control side, where significantly more plaque was found with more *Streptococcus*, *Acinetobacteria*, and *Capnocytophaga* species within the plaque. There was no significant rise in the MMP-8 levels in the GCF, despite more expression of MMP-8 on the control side. This might be partly due to the dilution of MMP-8 within the GCF fluid by sample collection. To exclude the possibility of such measuring technique-related distortion, we conducted a sample collection twice during the last day. Thus, at the end of the study, on Day 4, we collected GCF before and after the treatment session. This experiment showed a dilution phenomenon in the serially collected MMP-8 samples. Thus, the daily collection of GCF causes dilution, and the newly developed MMP-8 is not expressed at the same pace as it is lost in the samples. Thus, the measuring itself distorts the results, but, because this distortion is similar in all samples, the results are comparable to each other. This explains the reduction of MMP-8 concentration on both treatment and control sides during the first three days, and the slight increase on the fourth day. Hence, taking into consideration the levels of the MMP-8 enzyme and the dilution of the enzyme due to daily GCF collection, we decided to also compare the total amount of MMP-8 expressed (integral) between the treated and control sides. This showed significantly less MMP-8 expression on the aPDT-treated side. Whether the reduction in MMP-8 expression is due to the smaller amount of dental plaque, the change of bacterial flora, or both, remains obscure. Alternatively, the ROS induced can regulate, i.e., also oxidatively inactivate, MMP-8 [40,41].

Elimination of bacterial plaque from tooth surfaces is essential for maintaining good dental health [42,43]. With respect to the great magnitude of the unsatisfactory general state of dental hygiene and its consequences for health in the general population, new advancements are welcome. For home use, PDT could be an auxiliary method used to fight against biofilm infections. Mechanical cleaning is the most reliable treatment for biofilm infections, but, despite advances in communication and general knowledge regarding oral hygiene, the amount of dental and periodontal infections remains high. Mouthwashes have widely been used as a supplemental treatment modality. However, despite the antibacterial effect of the most popular mouthwashes, such as Listerine^®^, their effect on treating mouth infections remains unclear. Chlorhexidine, an effective and frequently used mouthwash, has side effects which hinder long-term usability, such as tooth staining, alterations in taste perception, and cross-resistance formation against colistin, a last resource antibiotic [44]. Antibacterial PDT, on the other hand, has multiple attributes, of which the most attractive is the lack of resistance formation, which would enable the use of the treatment repeatedly over the long term. The treatment also has the ability to whiten teeth, a highly desirable side effect [45]. The efficacy against biofilm bacteria is dose-dependent but also varies among different PDT light–photosensitizer combinations. Thus, aPDT could be used instead of or in conjunction with mouth rinses to prevent tooth decay or periodontitis.

The recent development of LED semiconductors has brought medical light technology ever closer to home use. In contrast to the coherent and intense laser beam used in laser-based devices, the light in LED light sources produces scattered, wide-spreading light, without hotspots. The light produced by LEDs reduces the risks associated with laser light, such as eye damage and thermal injury to tissues. Thermal injury in dentin tissue by laser light might build up at the dentino-enamel junction due to sharp change in light scattering at the border of dentin and enamel, which can cause energy accumulation there [46]. An additional benefit of LED-based light production is the improved total amount of light when compared to laser devices. Finally, the most enticing element of LED light is the ease of use and general safety, which is important for public acceptance if novel solutions are applied for general use.

## 5. Conclusions

Antimicrobial photodynamic therapy provided by near-infrared light and an ICG photosensitizer prevents plaque formation and selectively alters bacterial flora within the plaque. These two methods, used in combination, can help to reduce early gingivitis. Antimicrobial PDT offers an additional method for prophylactic dental hygiene.

## Figures and Tables

**Figure 1 dentistry-09-00052-f001:**
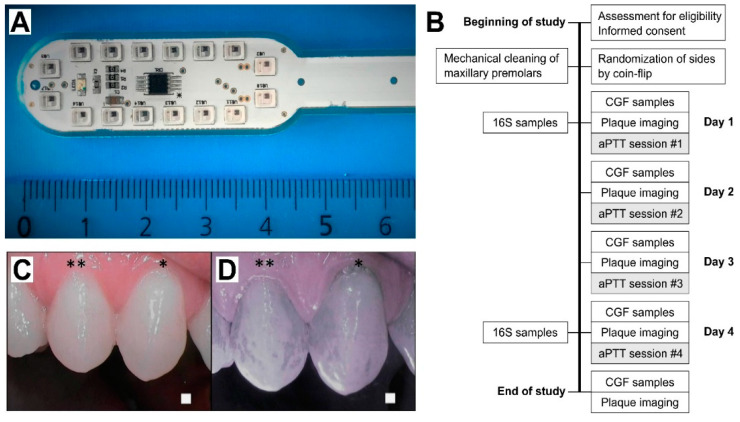
(**A**) LED light applicator; and (**B**) study workflow. 16S, bacteriome 16S rRNA sequencing; CGF cervical gingival fluid; aPDT, antimicrobial photodynamic therapy. (**C**,**D**) Selective indocyanine green (ICG) localization to the dental plaque. Daylight (**C**,**D**) near-infrared light images after ICG mouth rinse. Representative images at Day 2, when the teeth had been without cleaning for a single day. * premolar one and ** premolar two.

**Figure 2 dentistry-09-00052-f002:**
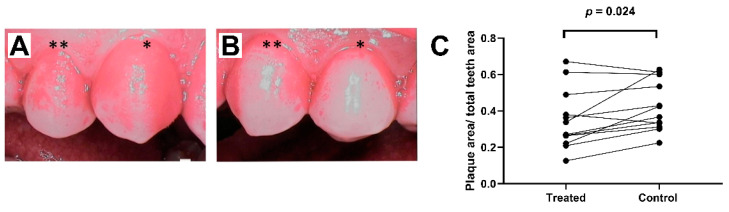
Dental plaque formation: (**A**) plaque areas (62.7% of total premolar dental area) on the control side in the last imaging session (premolar one (*) and premolar two (**)); (**B**) plaque areas (33.8% of total premolar dental area) on the treatment side at the same time point; and (**C**) plaque areas at the end of study after four days of aPDT application. Paired measurements demonstrated significantly less plaque formation on the treated side compared to control side.

**Figure 3 dentistry-09-00052-f003:**
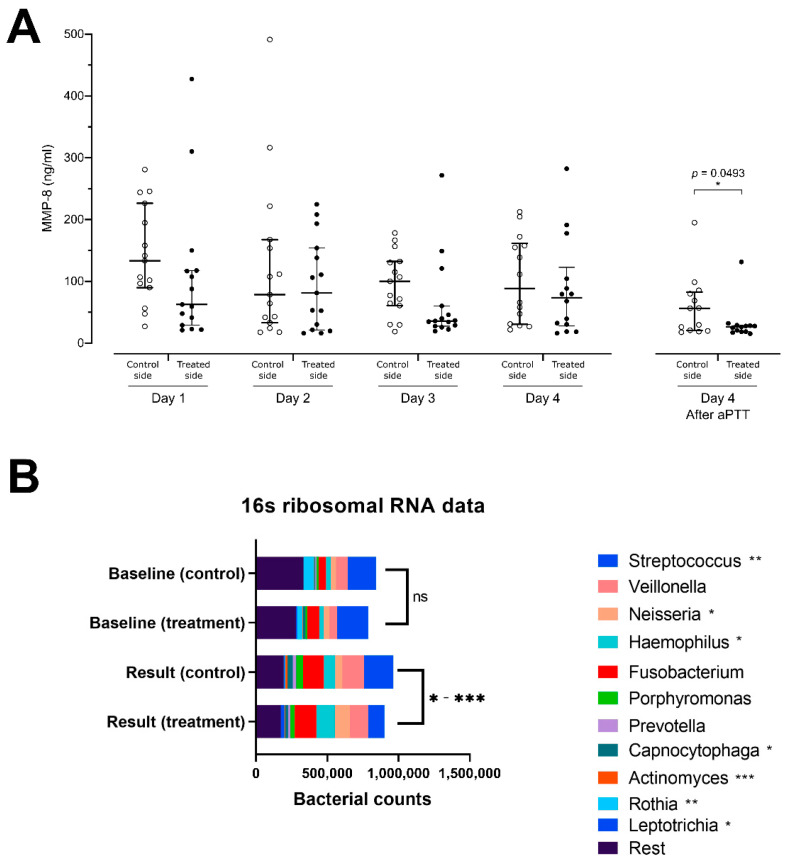
(**A**) Matrix metalloproteinase-8 (MMP-8) concentrations at indicated days and sides in samples from gingival pockets. aPDT effect on MMP-8 secretion significantly lower on the aPDT-treated side than on the control side after the last application of aPDT at end-of-study; and (**B**) analysis of bacteriome using 16S rRNA sequencing. This figure shows the diversity of bacteria within the plaque samples from the treatment and control premolars at the start of the treatment at baseline and at the end of the study period. At the end of treatment, a significant reduction in the relative proportion of *Streptococcus*, *Actinomyces*, and *Rothia* bacteria species was identified, and a relative increase in the *Neisseria*, *Haemophilus*, and *Leptotrichia* bacteria species was seen between treated and control side. * *p* < 0.05, ** *p* < 0.01, *** *p* < 0.001.

## Data Availability

The 16S rRNA sequences have been deposited to the Sequence Read Archive (SRA) of the National Center for Biotechnology Information (NCBI) under BioProject ID PRJNA661546.

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
