# Peer review of "Indocyanine Green-Assisted and LED-Light-Activated Antibacterial Photodynamic Therapy Reduces Dental Plaque"

_dentistry, 2021, doi:10.3390/dj9050052_

Round 1

Reviewer 1 Report

It is a well conducted study.

Some doubts remain about the small sample size and the aim of the study.

It would be interesting to compare the aPDT with the home mechanical therapy to evaluate the difference in terms of MMP-8 and plaque bacteriomes.  

After only 4 days plaque is characterized mainly by species responsible for dental caries. Subsequently dental plaque is colonized by anaerobic species responsible for periodontal diseases.  

The objective of the study is confounding. The Authors would like to demonstrate that ICG-assisted aPDT is effective in plaque control. It seems difficult for the moment to use this therapy day by day, compared with common oral hygiene maneuvers, for feasibility and costs.  

Author Response

We kindly thank the Editor and the reviewers for the constructive comments. We are grateful for your time and efforts to help revise our manuscript. We found the advice and thoughts clinically relevant and clearly improving the paper.

We have addressed all comments and criticism to the best of our ability. We hope that our answers satisfy all issues regarding the manuscript, however we are willing to make additional changes if they are necessary.

Changes in the revised manuscript are shown with yellow highlight in the Microsoft Word document. Please find our point-by-point responses below.

REVIEWER #1

It is a well conducted study

R1/Q1: Some doubts remain about the small sample size and the aim of the study.

A1: The sample size is small, but based on power calculations, now added in the Materials and Methods/statistical analysis-section (page 3, lines 141-145):

”Sample size calculations were carried out with the SAS 9.4 software (SAS Institute, Cary, NC, USA) power-procedure’s twosamplewilcoxon-statement using an alpha-error level of 0.05 to attain 90% statistical power. The calculated number of participants needed in the split mouth was 10. Eventually, we decided to enroll 15 participants for possible dropouts.

We wanted deliberately to use a very robust test, but keep the methods solid to be able to trust the observed results. The aim was to study feasibility as well as to gain first insight into efficacy.

R1/Q2: It would be interesting to compare the aPDT with the home mechanical therapy to evaluate the difference in terms of MMP-8 and plaque bacteriomes.  

A2: This is a very good suggestion. Actually, the MMP-8 level in each patient at the beginning of the study can be considered to reflect their regular mechanical oral care. We agree with the reviewer, that more research is needed to further evaluate the benefits of aPDT. It would, indeed, be interesting to compare aPDT with home mechanical care, perhaps as part of a larger clinical trial that also evaluates aPDT as an additive to home mechanical therapy. This could even be a longer-term clinical study. We consider, however, that mechanical treatment would beat the aPDT. Thus, we prefer to think that aPDT can serve as adjunctive therapy to the home mechanical oral care.

R1/Q3: After only 4 days plaque is characterized mainly by species responsible for dental caries. Subsequently dental plaque is colonized by anaerobic species responsible for periodontal diseases.  

A3: This is totally true. Furthermore, the studied population was healthy adults, without known periodontitis. We have added in the Discussion (page 6, lines 224-225):

“During the four days of development, the species in the plaque can be considered to generally consist of cariogenic bacteria, especially in the healthy population studied.”

R1/Q4: The objective of the study is confounding. The Authors would like to demonstrate that ICG-assisted aPDT is effective in plaque control. It seems difficult for the moment to use this therapy day by day, compared with common oral hygiene maneuvers, for feasibility and costs.  

A4: It is true that further development of the approach presented in this study is required to improve feasibility and to reduce costs. In fact, we have revecently developed a dual light antibacterial PDT treatment device system consisting of two CE-approved devices a ICG mouthrinse and a light guard providing antibacterial treatment in addition to mechanical home hygiene. The device cost is 249 € and the continuous use of the mouthrinse is about 1€/ use. You can find the device at www.lumoral.com.

Reviewer 2 Report

Dear Professor Jaquiéry,

Thank you for the opportunity to review the manuscript ID: dentistry-1146990 entitled "Indocyanine green-assisted and LED-light-activated antibacterial photodynamic therapy reduces dental plaque ".

The paper covered an interesting subject in dental biofilm, which is about investigating the effect of antimicrobial photodynamic therapy in the development of dental plaque. The study was well designed, and the results were discussed thoroughly, however, I have minor comments to the authors:

1- Line 15: it seems that the authors have included the aim under the background subheading, which doesn’t reflect the title of “background”
.

2- Lines 82-88: can the authors clarify the gender and age of these participants who completed the study (average and range) and whether there were any specific inclusion/exclusion criteria such as caries and periodontal status, smoker/non-smoker, receiving antibiotics in the last 3 months prior to the start of the study. 

In addition, can the authors confirm that all the participants had normal saliva secretion at the beginning of the trial?

3- Line 139: the authors should provide more information in the statistical analysis, such as the sample size calculation and the p-value that was considered statistically significant. 

4- Line 178: can the authors describe the calculation of AUC and include some information regarding this measure under “statistical analysis”.

5- The discussion was long; however, it was well written and interesting to read. I really appreciate the authors' analysis and reflection.

Thank you

Best wishes

Author Response

We kindly thank the Editor and the reviewers for the constructive comments. We are grateful for your time and efforts to help revise our manuscript. We found the advice and thoughts clinically relevant and clearly improving the paper.

We have addressed all comments and criticism to the best of our ability. We hope that our answers satisfy all issues regarding the manuscript, however we are willing to make additional changes if they are necessary.

Changes in the revised manuscript are shown with yellow highlight in the Microsoft Word document. Please find our point-by-point responses below.

REVIEWER #2

Thank you for the opportunity to review the manuscript ID: dentistry-1146990 entitled "Indocyanine green-assisted and LED-light-activated antibacterial photodynamic therapy reduces dental plaque ". The paper covered an interesting subject in dental biofilm, which is about investigating the effect of antimicrobial photodynamic therapy in the development of dental plaque. The study was well designed, and the results were discussed thoroughly, however, I have minor comments to the authors:

R2/Q1:- Line 15: it seems that the authors have included the aim under the background subheading, which doesn’t reflect the title of “background”
.

A1: We have changed the title of this subsection in the Abstract to “Aim” (page 1, line 15)

R2/Q2- Lines 82-88: can the authors clarify the gender and age of these participants who completed the study (average and range) and whether there were any specific inclusion/exclusion criteria such as caries and periodontal status, smoker/non-smoker, receiving antibiotics in the last 3 months prior to the start of the study. 

In addition, can the authors confirm that all the participants had normal saliva secretion at the beginning of the trial?

A1: This is a great question, All participants complied to the inclusion and exclusion criteria and all were healthy. We did not collect the DMFT data, but all the participants were periodontitis-free. We have added this information in the methods section (page 2, lines 82-90):

“Inclusion criteria included being generally healthy, being between the ages of 18 to 65, and having the ability to refrain from brushing one’s teeth during the treatment period. Exclusion criteria were diabetes, medications that may affect the immune response or saliva secretion, malignancy, pregnancy, dental implants or prosthesis, fixed orthodontic appliances, or active or chronic oral infection including gingivitis or active caries. Medications that could affect the oral microbiome, such as antibiotics, and antimicrobial mouthwashes, were not permitted during the study or within the two weeks prior to the study. Decayed, Missing, and Filled Teeth (DMFT) indexes were not collected.”

And to the results section (page 4, lines 157-160):

“All the subjects complied with the inclusion and exclusion criteria. The age of the subjects ranged from 23 to 48, with a mean of 27. Seven were female and eight males. All the study subjects were non-smokers. The study subjects were healthy, had no history of deranged saliva secretion, and were periodontitis-free.”

R2/Q3: 3- Line 139: the authors should provide more information in the statistical analysis, such as the sample size calculation and the p-value that was considered statistically significant. 

A3: We have added in the methods section the assessment of sample size (page 3, lines 141-145): “Sample size calculations were carried out with the SAS 9.4 software (SAS Institute, Cary, NC, USA) power-procedure’s twosamplewilcoxon-statement using an alpha-error level of 0.05 to attain 90% statistical power. The calculated number of participants needed in the split mouth was 10. Eventually, we decided to enroll 15 participants for possible dropouts. ”

We have also added to the Methods section (page 4, lines 150-151):

“A p value less than 0.05 was considered statistically significant.”

R2/Q4: 4- Line 178: can the authors describe the calculation of AUC and include some information regarding this measure under “statistical analysis”.

A4: Please see above answer A3.

R2/Q5: 5- The discussion was long; however, it was well written and interesting to read. I really appreciate the authors' analysis and reflection.

A5: We thank the Reviewer for the constructive comment. We considered the length of the Discussion-section carefully, and decided to keep the lengthy format due to reasons of providing the readers with enough insight to interpret the study results. Moreover, we believe this choice to be in-line also with the response of the Reviewer #3 applauding the writing of the manuscript.

Reviewer 3 Report

Article is very well written. 

Please add in the statistical analysis at which value You considered significant (the p value) 

the first paragraphs of the discussion need to be supported by references. 

Author Response

We kindly thank the Editor and the reviewers for the constructive comments. We are grateful for your time and efforts to help revise our manuscript. We found the advice and thoughts clinically relevant and clearly improving the paper.

We have addressed all comments and criticism to the best of our ability. We hope that our answers satisfy all issues regarding the manuscript, however we are willing to make additional changes if they are necessary.

Changes in the revised manuscript are shown with yellow highlight in the Microsoft Word document. Please find our point-by-point responses below.

REVIEWER #3

Article is very well written. 

R3/Q1 Please add in the statistical analysis at which value You considered significant (the p value) the first paragraphs of the discussion need to be supported by references. 

A1: We have added this information in the methods section (page 4, lines 150-151):  “A p value less than 0.05 was considered statistically significant.”

According to the Reviewer’s suggestion, we have revised the references in the first part of the Discussion-section. We have added the following new references to the revised manuscript:

- Marsh P, D: Dental Plaque as a Microbial Biofilm. Caries Res 2004;38:204-211. doi: 10.1159/000077756

- Odanaka, H., Obama, T., Sawada, N. et al. Comparison of protein profiles of the pellicle, gingival crevicular fluid, and saliva: possible origin of pellicle proteins. Biol Res 53, 3 (2020). https://doi.org/10.1186/s40659-020-0271-2